# Unrecognised water limitation is a main source of uncertainty for models of terrestrial photosynthesis

Samantha Biegel<sup>1,2</sup>, Konrad Schindler<sup>1,2</sup>, and Benjamin D. Stocker<sup>3,4</sup>

- <sup>1</sup> Photogrammetry and Remote Sensing, ETH Zürich
- <sup>2</sup> ETH AI Center, ETH Zürich
- <sup>3</sup> Institute of Geography, University of Bern
- <sup>4</sup> Oeschger Centre for Climate Change Research, University of Bern

**Correspondence:** Samantha Biegel (samantha.biegel@inf.ethz.ch)

Abstract. Quantification of environmental controls on ecosystem photosynthesis is essential to understand the impacts of climate change and extreme events on the carbon cycle and the provisioning of ecosystem services. Machine learning models have become popular for simulating ecosystem terrestrial photosynthesis because of their predictive skill, but often do not consider temporal dependencies in the data, even though process understanding suggests that these should exist. Here, we investigate how models that account for temporal structure impact the prediction of ecosystem photosynthesis. Using timeseries measurements of ecosystem fluxes paired with measurements of meteorological variables from a network of globally distributed sites (N = 104) and remotely sensed vegetation indices, we train three different models to predict ecosystem gross primary production (GPP): a mechanistic, theory-based photosynthesis model, a memoryless multilayer perceptron (MLP) and a recurrent neural network (Long Short-Term Memory, LSTM). Through comparisons of patterns in model error, we assess the ability of these models to predict GPP across a wide diversity of ecosystems and climates, and to account for temporal dependencies, with a focus on effects by low rooting zone moisture and freezing air temperatures. While both deep learning models outperform the mechanistic model, we find their overall performance is similar, with an  $R^2$  of 0.79 spatial out-of-sample predictions for both models. Overall, model skill is consistently good across moist sites with strong seasonality. During periods affected by temporal patterns such as drought and frost, the LSTM shows lower model error than the MLP as well as an LSTM with shuffled input, showing that there is an advantage from learned temporal dependencies. Generalisation patterns reveal that the LSTM tends to be more successful than the (time-agnostic) MLP in simulating GPP in dry environments. However, a large variability in model skill across relatively dry sites remained. This was not resolved by the inclusion of additional earth observation data, although this improved overall model performance. Insufficient information on the exposure and response to water stress and related effects on GPP appear to be dominant sources of error for modelling ecosystem fluxes across the globe. With the increasing frequency of hydroclimatic extreme events, effects of water limitation are expected to become more prevalent, which calls for models that better represent its impact on ecosystem function.

#### 1 Introduction

35

45

Photosynthesis plays a major role in the global carbon cycle and drives important ecosystem functions (Beer et al., 2010). Ecosystem-level gross  $CO_2$  uptake through photosynthesis is referred to as gross primary production (GPP) and varies in response to the environment. Understanding its variations across space and time as well as its dependencies on environmental conditions is key for predicting changes and feedbacks in the terrestrial biosphere (Booth et al., 2012).

GPP variations are driven by multiple, simultaneously varying environmental factors and the physiological and structural responses of plants to these conditions. Solar radiation supplies the energy for photosynthesis and acts as a dominant driver of GPP, depending on light absorption (Monteith, 1972). Temperature, light and water availability trigger phenological changes and regulate seasonal cycles of active leaf surface area and therefore seasonal changes in light absorption. Air temperature affects leaf temperatures, which in turn govern enzymatic rates and photosynthesis (Berry and Bjorkman, 1980; Kattge and Knorr, 2007; Kumarathunge et al., 2019; Bernacchi et al., 2003). Low moisture availability across the rooting zone, in combination with a high vapour pressure of air at the leaf surface, determines the effects of water stress and can lead to GPP reductions (Stocker et al., 2018; Novick et al., 2016).

Continuous GPP estimates can be obtained from eddy covariance measurements of ecosystem gas exchange (Baldocchi, 2020) and capture surface-atmosphere exchange fluxes, integrated over a radius on the order of a kilometre around the site of measurement (Chu et al., 2021). These measurements, paired with observations of meteorological variables and soil conditions, are made available through different networks and initiatives (e.g., AmeriFlux, ICOS, OzFlux). The combination of data from multiple regional networks has led to large datasets with standardized processing of eddy covariance measurements from a large number of sites (Pastorello et al., 2020; Hufkens and Stocker, 2025; Abramowitz et al., 2024). The availability of large datasets of GPP along with their environmental covariates, and paired with remotely sensed variables, has made machine learning (ML) a widely used approach for predicting spatio-temporal variations of ecosystem-atmosphere exchange fluxes (Kang et al., 2023; Gaber et al., 2024; Tramontana et al., 2016; Montero et al., 2024; Papale et al., 2015; Yang et al., 2007; Jung et al., 2011; Joiner and Yoshida, 2020; Zheng et al., 2020).

Process understanding and empirical patterns of GPP dynamics suggest that there should be temporal dependencies in data of GPP and its predictors. Temporal dependencies arise as a result of several processes. Low soil moisture can reduce GPP (Stocker et al., 2018) and reflects the history of precipitation, radiation, and leaf phenology over the preceding weeks to months. Plant hydraulic processes induce a temporal hysteresis effect over the course of diurnal cycles (Tuzet et al., 2003). Physiological changes are caused, e.g., by the seasonal acclimation of the photosynthetic apparatus to varying levels of radiation inputs and temperature (Kumarathunge et al., 2019; Luo and Keenan, 2020; Liu et al., 2024b; Berry and Bjorkman, 1980). Ecosystems in cold climates have been found to delay springtime photosynthesis resumption early in the season through photoprotective processes, despite high levels of solar radiation (Luo et al., 2023, henceforth referred to as "cold acclimation"). Stress by extreme environmental conditions can cause delayed and long-lasting effects, such as impaired transpiration and reduced CO<sub>2</sub> assimilation (Barber and Andersson, 1992; Reichstein et al., 2013; McDowell et al., 2022; Bastos et al., 2020; Yu et al., 2022).

Several published machine learning models for GPP treat values of GPP time series as independent and identically distributed observations and therefore do not account for temporal dependencies (Nelson et al., 2024; Kang et al., 2023; Tramontana et al., 2016; Gaber et al., 2024). This limitation may be relieved by temporal aggregation to daily-monthly time scales and by pairing data with additional, remotely sensed observations that capture phenological changes and variations in the amount of active, light-intercepting foliage area (Baldocchi, 2018). However, additional physiological changes that affect the efficiency of light utilization for CO<sub>2</sub> assimilation at the leaf level are more challenging to capture by remotely sensed reflectance data (Ryu et al., 2019; Stocker et al., 2018). As a consequence, substantial unexplained GPP variation is expected to remain at the seasonal and diurnal time scales.

55

A potential solution for this problem is the use of time-aware machine learning algorithms that can learn non-stationary relationships and temporal dependencies. Such algorithms have been introduced for modelling GPP and related fluxes (Nakagawa et al., 2023; Besnard et al., 2019; Kraft et al., 2024; Montero et al., 2024). Montero et al. (2024) compared the performance of three recurrent architectures for GPP modelling and evaluated them on GPP extremes. Besnard et al. (2019) evaluated a Long Short-Term Memory (LSTM) network to study net ecosystem CO<sub>2</sub> exchange. Kraft et al. (2024) assessed sequential models for global upscaling of evapotranspiration. Nakagawa et al. (2023) introduced a temporal fusion transformer for global upscaling of GPP. In these previous studies, the impact of using such an architecture for modelling known temporal effects was either not evaluated or inconclusive.

In contrast to ML models, mechanistic GPP models, such as the P-model (Stocker et al., 2020), embody process understanding and provide a theory-based prediction that accounts for these known temporal dependencies. The foundation in plant physiology may also help these models to generalise more robustly, as the underlying relations remain valid when extrapolating to new conditions not seen in the training data. The price to pay is that these models lack the flexibility to pick up any patterns that were not anticipated during their design, whereas the high representation power of ML models gives them the ability to uncover and respect such patterns.

While the P-model only uses a single greenness index derived from remote sensing, previous work on flux modelling has shown that additional remotely sensed variables can be informative (Nelson et al., 2024; Kraft et al., 2024). The signal from thermal remote sensing may reflect changes in photosynthesis that are driven by physiological responses and stomatal regulation, affecting transpiration and therefore surface energy partitioning and surface heating. Therefore, land surface temperature (LST) may be useful information for GPP prediction. Common mechanistic and light use efficiency models don't consider this information as additional forcing. Furthermore, the full information of surface reflectance in all available individual bands may contain additional information about GPP changes as pigments deployed under stress conditions or the leaf water content can affect surface reflectance beyond what commonly used single greenness-based indices reflect (Ceccato et al., 2001; Gamon et al., 2016).

In this study, we evaluate the use of an LSTM (Hochreiter and Schmidhuber, 1997) as a predictor of GPP. LSTMs have been shown to be successful at tasks where memory effects across a range of temporal scales are involved, such as sea surface temperature prediction (Zhang et al., 2017), rainfall-runoff modelling (Kratzert et al., 2018) and canopy greenness modelling (Liu et al., 2024a). To contrast the recurrent and purely data-driven design, we compare against a standard, non-recurrent multilayer

perceptron (MLP), an LSTM with permuted input, as well as to the process-based P-model. The P-model serves as a benchmark with known treatment of temporal effects. We compare these models based on model performance and generalisation capabilities. To investigate the ability of the LSTM to account for temporal dependencies, we assess seasonal patterns of cold acclimation and water limitation effects in dry conditions. Additionally, we analyse spatial patterns of model generalisability (spatial out-of-sample performance) with respect to different environmental factors.

In addition to the standard set of predictors, we include LST and multiple bands of surface reflectance. To aid the models in simulating temporal dependencies, we also provide additional features related to soil water availability. In view of the known influence of root zone moisture on GPP (Stocker et al., 2018) and the inability of the MLP to account for the precipitation and radiation history (and thus implicitly for the evolution of root zone moisture), we test if its performance improves when (simulated) soil moisture is provided as a complementary predictor. Providing observation-derived soil moisture or a general index of water availability as an additional predictor is a common approach taken also for other memoryless GPP models (Nelson et al., 2024; Kang et al., 2023; Tramontana et al., 2016; Gaber et al., 2024). While the LSTM is expected to be able to learn the effects of soil moisture limitation, we also test whether it benefits from an Earth-observation derived estimate of root zone water storage capacity (Stocker et al., 2023) as additional time-invariant context.

#### 2 Materials and methods

#### 105 2.1 Data

We sourced daily GPP data from a collection of eddy covariance flux sites gathered from the PLUMBER2 framework (Ukkola et al., 2022), which includes sites from OzFlux (Isaac et al., 2017), FLUXNET2015 (Pastorello et al., 2020) and LaThuile; as well as AmeriFlux, ICOS Warm Winter 2020 (Warm Winter 2020 Team and ICOS Ecosystem Thematic Centre, 2022) and ICOS Drought 2018 (Drought 2018 Team and ICOS Ecosystem Thematic Centre, 2020). Site selection was performed through several steps. Sites located in cropland or wetland ecosystems were excluded. For each site, we only included full calendar years of data. Some years of data were excluded due to evident inconsistencies found by visual inspection. We selected sites with at least five consecutive years of high-quality, gap-free data. GPP data were included if at least 50% of all half-hourly measurements were of good quality (either measured or gap-filled with high confidence).

We used GPP estimates generated by the nighttime partitioning method (GPP\_NT\_VUT\_REF; Reichstein et al., 2005). Half-hourly GPP estimates were aggregated to obtain daily GPP values. In addition to GPP, meteorological variables were obtained, which were measured directly at the flux sites. We used the following meteorological variables: air temperature (TA\_F\_MDS), daytime air temperature (TA\_DAY\_F\_MDS), shortwave incoming radiation (SW\_IN\_F\_MDS), longwave incoming radiation (LW\_IN\_F\_MDS), daytime vapour pressure deficit (VPD\_DAY\_F\_MDS), air pressure (PA\_F), precipitation (P\_F) and wind speed (WS\_F). Observations that were either missing or had insufficient quality (

**Figure 1.** Site locations and their moisture indices. For each category, the quantised moisture index values and number of sites are given in brackets.

Along with local site-level measurements, we used remotely sensed estimates of the fraction of absorbed photosynthetically active radiation (fAPAR), extracted from the MODIS FPAR MCD15A2H Collection 6.1 product (Myneni et al., 2021). fAPAR captures variations in phenology and represents the amount of solar radiation absorbed by the canopy and usable for photosynthesis. fAPAR data were extracted for the pixel  $(500 \times 500 \,\mathrm{m}^2)$  area) that contains the flux measurement site and for the eight pixels immediately surrounding it. The nine values where combined through a weighted average, using as weights the inverse of their variance as per the data product. The fAPAR sequences were gap-filled based on the mean seasonal cycle, then smoothed and interpolated to the time resolution of the flux data with a LOESS spline.

We retrieved additional pre-processed remote sensing variables from the MODIS instruments through the FluxnetEO dataset (Walther et al., 2022). We included the first seven reflectance bands (RED, NIR, BLUE, GREEN, SWIR1, SWIR2, SWIR3) as well as daytime and nighttime land surface temperature (LST) at a viewing zenith angle of 0° (LST\_TERRA\_Day\_VZA0, LST\_TERRA\_Night\_VZA0).

The collection of flux sites was further filtered to include only sites with data available from the FluxnetEO dataset. This process resulted in a collection of 104 sites (Figure 1) with a total of 1020 site-years of data. Detailed site information is given in Appendix A. The selected sites cover a wide range of environmental factors, in particular aridity (Figure 1 and Figure D1). To measure aridity, we calculated the moisture index (MI) for each site as total P/PET, where P is the precipitation measured at the site and PET is the potential evapotranspiration following Priestley-Taylor (Priestley and Taylor, 1972), as implemented in the SPLASH ecosystem water balance model (Davis et al., 2017). The root zone water holding capacity for each site was sourced from Stocker et al. (2023).

#### 140 **2.2 Models**

135

We have implemented two deep learning models to evaluate different architectures for the prediction task. To account for temporal dependencies, we used a Long Short-Term Memory (LSTM) network (Hochreiter and Schmidhuber, 1997). The

network included LSTM cells with layer normalization, known to stabilize hidden dynamics and reduce training time (Ba et al., 2016). The number of layers and hidden dimension of the network were tuned as hyperparameters on validation data. The LSTM layers were followed by a variable number of linear layers with GELU activations (Hendrycks and Gimpel, 2023) that each halve the dimension until reaching a size of 16 neurons.

To isolate the impact of recurrence, we implemented a second neural network model without any memory mechanism, namely a standard multilayer perceptron (MLP). Its architecture is identical to keeping only the linear layers of the LSTM. The hidden dimension of the first layer was tuned as a hyperparameter. The architectural similarity means that the difference between predictions of the two networks is a good indication for the influence of information from past time steps. Hyperparameters were tuned separately to ensure both variants reach their best performance. We also evaluated a version of the LSTM where we permute the temporal ordering (LSTM<sub>perm</sub>). This model has the same architecture and capacity but it cannot rely on memory as the temporal patterns are eliminated from the input data. This version serves as an additional evaluation that disentangles the influence of the memory of the LSTM from other model properties such as the overall capacity.

The third model was the P-model, a mechanistic, theory-based representation of ecosystem-level photosynthesis acclimation and GPP (Stocker et al., 2020; Wang et al., 2017; Prentice et al., 2014). It builds on the widely used Farquhar-von Caemmmerer-Berry (FvCB) model for leaf-level C<sub>3</sub> photosynthesis (Farquhar et al., 1980). The FvCB model is combined with an optimal balancing of the costs of carbon assimilation and transpiration (Prentice et al., 2014). Furthermore, the P-model implements the coordination hypothesis, which states that photosynthesis is balanced at the intersection of light and Rubisco-limited assimilation rates during average daytime conditions (Maire et al., 2012). Based on these relations, the P-model predicts photosynthesis acclimation parameters to describe the processes that determine the light use efficiency (LUE). GPP is then modelled as the product of LUE and absorbed photosynthetically active radiation (APAR), which in turn is taken to be the product of the photosynthetic photon flux density (PPFD) and fAPAR. The forcings for the P-model correspond to the input data of the LSTM and the MLP. We used the FULL model setup as described in Stocker et al. (2020), which includes an empirical soil moisture stress function and temperature dependency of the intrinsic quantum yield. These two components were calibrated to the data with an optimization of four parameters through minimising the root mean squared error with the generalised simulated annealing algorithm, as implemented in the GenSA R package (Xiang et al., 2013). The P-model is implemented in the R package rsofun (Stocker et al., 2024).

### 2.3 Experimental setup

150

155

175

We assessed the three models for their ability to handle temporal dependencies and their ability to generalise to new sites with different environmental conditions.

**Global model.** We first evaluate each model in a spatial cross-validation setup, which measures performance at sites that were not seen during model training (where "training" of the P-model means calibration). For the spatial cross-validation, we assigned each site to one of five folds, stratified based on the per-site mean air temperature and the moisture index to achieve a similar distribution of climate types in all folds. In turn, four folds served as training data to fit the model weights and tune the hyperparameters, then GPP predictions were produced for the test sites in the fifth, held-out fold.

**Site-specific model.** To separate changes in environmental conditions (which can be covered by the global model) from potential variations of the functional relationships between different sites (which cannot be represented by a single set of model parameters), we also fit separate per-site models and evaluate them with a temporal cross-validation. In that setup, the temporal sequences of predictors and GPP at every individual site are split into years, setting the start of the year to the coldest month for sites in temperate, continental and polar climates, and to the wettest month for tropical and arid sites (following the Köppen-Geiger climate code). Cross-validation then proceeds by holding out every year in turn, and fitting the model on the remaining years.

180

In both setups, the models are trained on chunks of 128 days, whereas testing was performed on the full sequences (i.e., all data of a site in the spatial cross-validation setup, respectively individual years in the temporal cross-validation setup). The training chunks were created per site, with a random start date of the initial chunk within the first 96 days and a sliding window with regular overlap of 32 days. The features were standardized using the mean and standard deviation of the training folds.

Models were trained by minimising the mean squared error with the Adam optimizer (Kingma and Ba, 2017). Training was performed for a maximum 50 iterations and stopped after 10 iterations without improvement in the validation loss. To limit overfitting, an L2 penalty was applied on the parameter updates, and dropout (Hinton et al., 2012) was used after each LSTM layer except for the last layer. The learning rate was adaptively reduced when the loss no longer improved for several iterations. The batch size, the weight of the L2 penalty, the dropout rate, the initial learning rate, the patience before reducing the learning rate as well as the reduction factor were all tuned in an inner cross-validation loop within each data fold. In the spatial cross-validation setup, a 3-fold inner cross-validation was performed within each of the 5 data folds. The sites were distributed based on mean air temperature and moisture index to ensure equal representation of climate types across folds. The hyperparameters were tuned using random sampling of 20 configurations. In the temporal cross-validation setup, hyperparameters were tuned for each held-out test year based on the remaining years of data for each site. A cross-validation with 5 randomly chosen held-out validation years was performed. In this setup, the hyperparameters were tuned using random sampling of 40 configurations. In both setups, the configuration with the lowest average validation RMSE across the held-out folds was selected. The set of options for each hyperparameter is listed in Table 1. After hyperparameter tuning, models were trained on the full data folds with the chosen hyperparameter configurations, with 20% of the sites used as validation data for early stopping of the training.

GPP predictions were evaluated using the squared Pearson's correlation coefficient  $(R^2)$  and root mean squared error (RMSE). In addition to assessing the daily predictions, we aggregated predictions and observations to different scales. We calculated the mean seasonal cycle by averaging over all years observed at a site to obtain a mean value per day of the year. Predictions and observations were also temporally aggregated to site-level means. Moreover we calculated daily anomalies, defined as deviations between the daily values from the mean seasonal cycle; as well as yearly anomalies, defined as deviations between a site's annual mean values and its global, multi-year mean.

The site-specific models were evaluated for test years that start at the wettest or coldest month. For days before the first day of the first such month, no predictions where made. When comparing the site-specific model and the global model at the site level, we therefore filter the predictions of the global model to comprise the exact same test days as the site-specific model.

When evaluating the P-model, the same spatial cross-validation was used as for the machine learning models, with the model parameters calibrated separately for each fold (Stocker et al., 2024).

We investigated the ability of the models to capture two different, well-known temporal effects. To test how well soil moisture effects are reproduced, we looked at the (absolute) percentage error of the model predictions as the potential cumulative water deficit (PCWD) increases. PCWD was calculated from the flux data as the cumulative difference between potential evapotranspiration (PET) and precipitation (Stocker, 2021). PET was estimated based on Priestley-Taylor (Priestley and Taylor, 1972), as implemented in the SPLASH ecosystem water balance model (Davis et al., 2017). Values were pooled from all sites and all test days per PCWD interval, with intervals chosen such that they have at least 100 data points. We evaluated the temporal effect from soil moisture from a different angle through an analysis of model bias during drought events identified from the full dataset. Drought events were collected through the identification of sites and days where light use efficiency was reduced for at least 3 days (Stocker et al., 2018). All events were aligned at the onset. The bias was calculated as the difference between model predictions and observations and aggregated per day for the period between 10 days before the start of each event and up until 100 days after the start of each event. The bias values were normalised by subtracting each value by the median value during the window covering between 1 and 10 days before each event.

Second, we evaluated the models' ability to reproduce cold acclimation effects. We selected four sites that have been found to have a reduced light use efficiency and thus a delayed increase in GPP at the start of the growing season (DE-Hai, US-Ha1, US-MMS, US-PFa). Prediction errors at these sites were contrasted with those at four sites that did not exhibit any GPP delay (BE-Vie, FI-Hyy, NL-Loo, RU-Fyo), based on the findings of Luo et al. (2023). For the two groups of sites, we aggregated and compared predictions per day of the year with different models.

For the global LSTM model, we compared site-level performance across various environmental conditions: moisture index (P/PET), Köppen-Geiger climate zone (Beck et al., 2018), and IGBP vegetation type (International Geosphere-Biosphere Programme). To further investigate the generalisation of the models across sites, we compared site-level performance between the global model and the site-specific model, by computing the relative difference  $\Delta R^2 = R_{\rm global}^2 - R_{\rm site}^2$  and the ratio  ${\rm rRMSE} = {\rm RMSE}_{\rm global}/{\rm RMSE}_{\rm site}$ .

We evaluated two different sets of predictor variables. The standard set included the site-level meteorological measurements as well as fAPAR. The expanded set included LST variables and the MODIS reflectance bands. The model evaluations with this expanded set are denoted by a + after the model name. We compared this setup to the standard predictor set in terms of overall performance. Separately to the standard setups, we fed additional predictors to the feature set of the deep learning models. For the MLP, we added soil moisture. Due to the limited quality of measured soil moisture at many flux sites, we used modelled soil moisture from the SPLASH water balance model (Davis et al., 2017). For the LSTM, we added the root zone water storage capacity, extracted from the global map of Stocker et al. (2023).

Finally, we evaluated a Temporal Convolutional Network (Bai et al., 2018) and an LSTM with attention layer (Vaswani et al., 2017) as two alternative sequence models (Appendix C). We compared their overall performance as well as their event response performance to the LSTM.

The details of the data, models and experimental setup are summarised in Table B1, Table B2 and Table B3.

| LSTM and MLP             |                                                                 |  |  |  |
|--------------------------|-----------------------------------------------------------------|--|--|--|
| Hidden dimension         | 64, 128, 256, 512                                               |  |  |  |
| Learning rate            | $10^{-2}$ , $5 \times 10^{-3}$ , $10^{-3}$ , $5 \times 10^{-4}$ |  |  |  |
| Scheduler patience       | 2, 3, 5                                                         |  |  |  |
| Scheduler factor         | 0.1, 0.5                                                        |  |  |  |
| Weight decay $(\lambda)$ | $10^{-3}, 10^{-4}, 10^{-5}, 0$                                  |  |  |  |
| Batch size               | 64, 128, 256                                                    |  |  |  |
| LSTM-only                |                                                                 |  |  |  |
| Dropout                  | 0, 0.1, 0.2, 0.3, 0.4                                           |  |  |  |
| Number of layers         | 1, 2, 3, 4, 5                                                   |  |  |  |

**Table 1.** Hyperparameter search space for the LSTM and MLP models.

#### 3 Results

## 3.1 Overall performance

Overall, we found that both machine learning models in the global setting (i.e., a single, fixed model trained on multiple sites) predict GPP more accurately than the process-based P-model (Table 2). Furthermore, the model variants with the additional remote sensing variables (+ version) performed better than the models with only the standard predictor set. For daily predictions, the  $R^2$  calculated from pooled data of all sites was 0.79 for both the LSTM+ and the MLP+, compared to 0.64 for the P-model. The RMSE was 22.5% lower for the LSTM+ than for the P-model. Both the LSTM+ and MLP+ modelled the seasonal cycle well, with  $R^2$  values of 0.88. The P-model achieved an  $R^2$  of 0.78 for modelling the mean seasonal cycle. The prediction of anomalies was more challenging for all three evaluated models. The LSTM+ achieved an  $R^2$  of 0.28 for daily anomalies and  $R^2$  of 0.1 for annual anomalies. While the differences in both  $R^2$  and RMSE were minimal between the LSTM+ and MLP+, the P-model was outperformed in both aspects.

Model performance varied substantially between sites (Figure 2). For 94 (out of 104) sites the LSTM+ reached a higher  $R^2$  value than the P-model. The LSTM+ also outperformed the MLP+ at slightly more than half of the sites. While the LSTM+ clearly outperformed the LSTM in terms of overall metrics, the performance didn't improve at every site. At 68 out of 104 sites, the LSTM+ outperformed the LSTM. In further analyses, we focus on the + versions of each model, as the overall performance evaluation indicates that they are better predictors of GPP.

More salient differences between the models were observed when inspecting the predicted mean seasonal cycles within different climate zones (Figure 3). The deep learning models were better at predicting the timing of early spring GPP increase in several climates (Köppen-Geiger codes Dfb, Dfc, Cfa). They also outperformed the P-model in desert and semi-arid climates.

| Metric | Aggregation  | LSTM | LSTM+ | LSTM <sub>perm</sub> | LSTM <sub>perm</sub> + | MLP  | MLP+ | P-model |
|--------|--------------|------|-------|----------------------|------------------------|------|------|---------|
| $R^2$  | Daily        | 0.75 | 0.79  | 0.74                 | 0.77                   | 0.75 | 0.79 | 0.64    |
|        | Seasonal     | 0.84 | 0.88  | 0.83                 | 0.87                   | 0.84 | 0.88 | 0.78    |
|        | Spatial      | 0.73 | 0.81  | 0.73                 | 0.77                   | 0.73 | 0.82 | 0.61    |
|        | Daily anom.  | 0.26 | 0.28  | 0.25                 | 0.26                   | 0.26 | 0.27 | 0.2     |
|        | Annual anom. | 0.1  | 0.1   | 0.14                 | 0.1                    | 0.12 | 0.12 | 0.05    |
| RMSE   | Daily        | 1.92 | 1.79  | 1.96                 | 1.85                   | 1.92 | 1.77 | 2.32    |
|        | Seasonal     | 1.37 | 1.2   | 1.4                  | 1.26                   | 1.39 | 1.17 | 1.65    |
|        | Spatial      | 0.89 | 0.74  | 0.89                 | 0.82                   | 0.89 | 0.71 | 1.06    |
|        | Daily anom.  | 1.37 | 1.35  | 1.39                 | 1.39                   | 1.37 | 1.37 | 1.64    |
|        | Annual anom. | 0.44 | 0.44  | 0.42                 | 0.44                   | 0.42 | 0.43 | 0.46    |

**Table 2.** Performance metrics ( $R^2$  and RMSE) of the global model at different aggregation levels. Metrics are calculated from pooled data of all sites. "Daily" evaluates daily predictions and observations. "Seasonal" aggregates by day of year per site. "Spatial" uses site means. "Daily anom." is the deviation from each site's mean seasonal cycle. "Annual anom." is the deviation of the annual mean from the multi-year mean per site. The best values per row are printed in bold.

Figure 2. Comparison of the LSTM+ against the P-model (a), against the MLP+ (b), and against the LSTM (c).  $\mathbb{R}^2$  of predicted versus observed values of daily GPP per site from the global cross-validation is shown for the LSTM+ along the y-axis and the P-model/MLP/LSTM along the x-axis. The dotted line indicates equal performance.

#### 265 3.2 Temporal patterns in model error

From the overall performance metrics, it appeared that the LSTM+ is not clearly better than the MLP+, despite its ability to learn temporal patterns. However, the improved performance of the LSTM+ compared to the LSTM<sub>perm</sub>+ indicates that temporal

**Figure 3.** Mean seasonal cycle of GPP and model predictions by climate zone and hemisphere. Predictions from the global cross-validation for the LSTM+, LSTM<sub>perm</sub>+, MLP+ and P-model are compared against GPP observations. Climate zone boundaries are from Beck et al. (2018).

dependencies were learned. Targeted evaluations of temporal patterns give a clearer insight into the differences between the MLP+, the LSTM<sub>perm</sub>+ and the LSTM+.

The LSTM+ showed different error characteristics than the P-model at high values of PCWD. While relative errors increased with higher PCWD for the P-model, they stayed relatively constant for the LSTM+ (Figure 4a.) For all pooled data, the four models showed similar error distributions at lower levels of PCWD, again the errors of the P-model increased at higher PCWD values, whereas they did not for the LSTM+, LSTM<sub>perm</sub>+ and the MLP+.

**Figure 4.** Error distribution across different amounts of cumulative water deficit (PCWD, quantized into 100 mm bins). Bins with less than 100 samples are not shown. Predictions are by the global models. Lines denote the median values, shaded regions lie between the lower (25%) and upper (75%) quartiles. (a) all data pooled. (b) Evergreen sites, including Evergreen Needleleaf Forest and Evergreen Broadleaf Forest. (c) Non-evergreen sites, including all other vegetation types (Deciduous Forests, Shrublands, Savannas and Grassland).

Separating the analysis of model errors versus PCWD by vegetation type revealed differences at higher PCWD values (Figure 4b.). For evergreen forests, relative errors increased for the P-model, the MLP+ and the LSTM<sub>perm</sub>+ from a PCWD of 800 mm, but also decreased again. The relative errors increased less for the LSTM+ than for the other models. For non-evergreen forests, both the LSTM+ and the MLP+ showed lower relative error than the P-model above a PCWD of 1000 mm.

The comparison between sites with and without cold acclimation (delayed GPP) revealed clear differences w.r.t. the predicted seasonal cycles (Figure 5) of the different models and the seasonal cycle of model bias. For sites without a delay in springtime GPP increase, the LSTM+ performed best during spring and summer. For sites with cold acclimation, the deep learning models capture the delay better than the P-model. The LSTM+ predicts the evolution of GPP best during springtime, although that edge is mostly after the start of spring, whereas some bias remains at the onset of GPP increase as well as during summer.

The event analysis (Figure 6) showed that the deep learning models predicted the drought response equally well during the first 20 days after an event. The LSTM+ predicted the response best from 20 days after the start of a drought event.

Evaluations of the alternative sequence models (Appendix C) showed that while the TCN+ performed better at 60 out of 109 sites (Figure C1), no difference was found between models in terms of their performance during drought events (Figure C2), supporting the further evaluation of the LSTM+ for temporal patterns.

#### 3.3 Spatial patterns in model performance

The observed error patterns suggest a qualitatively different behaviour of machine learning models, especially the LSTM+, during conditions where temporal effects are known to occur. To investigate this further at the site level, we plot model performance per site against relevant site characteristics (Figure 7).

## Sites with delayed GPP (n=4) Sites with non-delayed GPP (n=4)

**Figure 5.** Mean seasonal cycle of GPP and model predictions of the LSTM+, LSTM<sub>perm</sub>, MLP+ and P-model for four sites with delayed GPP (a) and four sites without delayed GPP (b), as well as mean bias per day of the year for sites with delayed GPP (c) and non-delayed GPP (d). The bias is calculated as the difference between model predictions and observations.

The LSTM+ performed best for relatively moist sites. Across sites with moisture index  $P/PET \ge 0.75$  the median  $R^2$  is 0.82, whereas it was only 0.65 for more arid sites (MI 

**Figure 6.** The model bias per day before and after the start of drought events. The bias is calculated as the difference between model predictions and observations. The bias is normalised with respect to the median value during the 1 to 10 days before each event onset. Shaded areas mark the area between the 33% and 66% quartiles.

All sites with low  $R^2$  (

Figure 7. Performance as  $R^2$  (a) and normalised RMSE (b) of the LSTM+ (global model) per site against different site characteristics. Section (a) and (b) each show comparisons of model performance against different site characteristics: moisture index, vegetation type and climate zone. The plots on the left show model performance per site against the moisture index. The plots on the right show model performance per site for different categories of vegetation type and climate zone, as well as box plots summarising the performance in each category. GRA: grassland, SAV: savanna, EBF: evergreen broadleaf forest, WSA: woody savanna, MF: mixed forest, CSH: closed shrubland, ENF: evergreen needleleaf forest, DBF: deciduous broadleaf forest, OSH: open shrubland.

semi-arid climates, generalisation capabilities vary substantially, with the global model outperforming the site-specific one in some sites but not in others.

Figure 8. Patterns in the LSTM+'s ability to generalise, measured by  $\Delta R^2$  (a) and rRMSE (b). Positive  $\Delta R^2$  values and rRMSE values 

Figure 9. Per-site comparison of  $\Delta R^2$  LSTM+ vs. MLP+. Colours encode the moisture index. The number of sites on either side of the diagonal and their mean moisture index are displayed in the corners. At the top we show the significance of the difference in MI, as per the two-sided Mann-Whitney U-test.

#### 3.5 Performance with additional features

Based on the observed temporal error patterns in response to water deficit, we tested whether the memoryless MLP+ would benefit from soil moisture as an added predictor (Figure 10). Including soil moisture information from the SPLASH water balance model led to a minimal difference in overall performance. There is a trend that adding soil moisture improves the MLP+ prediction at drier sites (mean MI 0.93), but leads to a small performance loss at moist sites (mean MI 1.10).

We also tested the LSTM+ with the estimated root zone water holding capacity as additional (time-invariant) predictor (Figure 10). While this led to minimal overall differences, the performance improved at both very moist and very dry sites.

#### 330 4 Discussion

## 4.1 Key insights

We have compared different model types for predicting GPP from a shared set of predictors. To summarise, the following key results were obtained:

- The neural networks (LSTM and MLP) are better GPP predictors than the mechanistic model (P-model) (Table 2, Figure 2, Figure 3).

Figure 10. Impact of complementary input features (soil moisture for the MLP+, water holding capacity for the LSTM+). (a) to (c) show the  $R^2$  and (d) to (f) show the RMSE. (a) and (d) compare the MLP+ with soil moisture to the MLP+. (b) and (e) compare the LSTM+ to the MLP+ with soil moisture. (c) and (f) compare the LSTM+ with water holding capacity to the LSTM+. Each dot represents the performance for one site. The number of sites on either side of the diagonal and their mean moisture index are displayed in the corners. At the top we show the significance of the observed differences, as per the two-sided Mann-Whitney U-test.

- The LSTM improves GPP prediction compared to the LSTM<sub>perm</sub> (and MLP) under temporal stress patterns by leveraging learned memory (Figure 4, Figure 5, Figure 6).
- When averaged across conditions, the LSTM is not clearly better than the MLP (Table 2). However, the LSTM shows advantages under drought conditions (Figure 4, Figure 6, Figure 9). This can be mitigated to some extent by providing a simple soil moisture index (Figure 10).
- With added earth observation variables, there is a clear improvement in model performance (Table 2, Figure 2). However, there is still a large variability in model performance in dry conditions (Figure 7, Figure 8).

In the following, we discuss these insights in more detail.

#### 4.2 Neural networks are skilled GPP simulators

An initial conclusion from our experiments is that neural network models have higher predictive skill than the theory-based P-model, across all levels of aggregation (Table 2). The main advantage of neural models is their capacity to represent complex

functional dependencies, including effects that may not have been anticipated when deriving a model from plant physiological theory. Importantly, the neural networks predict GPP more accurately at *unseen* test sites. In other words, learning does not overfit the specific data streams at the training sites but discovers transferable patterns that are valid across space, and thus implicitly across environmental gradients. We attribute this robustness to the diversity of sites in our dataset, and to careful (fully automatic and data-driven) hyperparameter tuning.

In contrast to the deep learning models (LSTM+ and MLP+), the P-model implements rigid functional dependencies derived from a simplified depiction of the underlying processes (e.g., the big-leaf representation of canopy light absorption and a schematic, empirical treatment of water stress effects). The model has only few parameters to calibrate to the data (in our case four). On the one hand, so few degrees of freedom deprive the model of the ability to adapt to small but persistent effects present in the data; making it less accurate. On the other hand, they prevent it from going too far astray in the face of unexpected inputs.

Considering overall evaluations across all sites and dates, the two neural models perform equally well. However, the advantage of the LSTM+ over the other models was clear under certain conditions that we expected from the outset to underlie temporal structure in the data - water stress (Figure 4) and frost/cold acclimation (Figure 5). Under these conditions, the LSTM+ outperforms both the MLP+ and also the mechanistic model. This indicates a lack of current mechanistic understanding of processes affecting GPP under these conditions.

It should also be noted that the collection of sites used here represents only a limited subset of all relevant environments on Earth. None of the sites are located in a tropical ever-wet climate and certain conditions and combinations of vegetation types and species, environments, and plant growth conditions (soil, subsurface hydrology) may not be covered by our spatial cross-validation setup. The limited data availability is exacerbated by the fact that seasonal variations in GPP and environmental conditions tend to be very small in the tropics. Hence, it remains unclear whether the learned models extrapolate to conditions not well represented in our data collection (Ludwig et al., 2023; Meyer and Pebesma, 2022). While it is technically possible to predict GPP wherever the input predictors are available, such upscaling should be done with caution, and the limited reliability under particular conditions should be considered. Our results suggest that predictions are least reliable in regions with pronounced seasonal or perennial water limitation.

Analysing patterns in prediction error of the different models and model performance of out-of-sample predictions has revealed several key insights for (data-driven) modelling of terrestrial photosynthesis and its limitations. In the following, we discuss them in more detail.

## 4.3 Modelling cumulative and lagged effects on GPP benefits from a recurrent model

The prediction errors for the MLP+, LSTM<sub>perm</sub>+ and P-model tended to grow with increasing water deficit, but errors of the LSTM+ remained smaller for moderate levels of water deficits compared to the other models (Figure 4). Event analysis also showed that the recurrent architecture helps predict the response in GPP to drought events (Figure 6). The benefit was observed primarily from 20 days after the start of a drought event. The similarity in prediction errors among models during the start of drought events and at lower levels of PCWD appears to indicate that the recurrent architecture only benefits the prediction

of GPP at higher levels of water stress. On the other hand, as the LSTM+ was trained with sequences of 128 days, it could adapt to water limited conditions that build up over a period of up to 4 months. Due to the limited sequence length, longer periods of water stress were presumably not learned, which may contribute to rising errors towards the high end of cumulative water deficits and the relatively poor prediction of annual anomalies which may be driven by variable hydroclimatic conditions across years.

In this context, we point out that several factors likely degrade the prediction of annual anomalies: Inter-annual variability of ecosystem fluxes likely reflects effects by specific site histories which are not reflected in the predictor variables (Abramowitz et al., 2024), from inconsistencies in measurements of fluxes and meteorological covariates across years (e.g., sensor replacements), or by lagged effects of climatic extreme events (Zscheischler et al., 2014). By their nature, such effects are difficult to learn from example data spanning at most a few decades.

Adding soil moisture as a predictor to compensate for the MLP+'s lack of memory did not clearly boost overall performance. This indicates that (simulated) soil moisture does not fully account for the effects of gradually changing water stress, calling into question a widely used practice (Nelson et al., 2024; Kang et al., 2023; Tramontana et al., 2016; Gaber et al., 2024) and likely relates to a general challenge in accurately modelling water stress effects, which we discuss in more detail below. Soil moisture information did, however, improve GPP prediction at relatively arid sites at the cost of a slight drop at moist sites – nudging the behaviour of the MLP+ towards that of the LSTM+. We speculate that this trade-off could hint at a dependence of the functional relationships on aridity. The dependence of GPP on a soil moisture optimum that shifts in response to the growing season soil moisture (Peng et al., 2024) could also contribute to an advantage of the LSTM+ compared to the MLP+ with the current value of soil moisture.

Conversely, the LSTM+ had a (small) disadvantage in moist regions compared to the MLP+ (Figure 9); while there was no obvious relation between the moisture index and the preference for global or site-specific modelling. Taken together, it seems that the LSTM+ more consistently generalises across different aridity levels than the non-recurrent model. This could be an indication that the functional relationships it uncovers hold over a wider range of aridity regimes. These patterns are unlikely to stem from uneven spatial representation, data sampling artifacts or differences in training data length. These were accounted for by ensuring even representation and sequence length of moist and arid sites (Figure D1) as well as even distribution of moisture index values when splitting sites during training. Therefore, we attribute performance gaps to the ability of the models to account for the ecological complexity under arid conditions.

The LSTM+ also better captured delayed GPP increase in spring due to the cold acclimation effect (Figure 5). Luo et al. (2023) found that a reduced efficiency of photosynthetic light utilisation during springtime was a consequence of a combination of low minimum temperatures and high radiation during the weeks and months leading up to and during the start of the growing season. A recurrent deep learning model offers a basis for more accurately modelling GPP under such conditions than non-recurrent architectures. Unresolved challenges remain, though, in the form of a remaining marked bias in the early part of spring also for the LSTM+.

Overall, we found that the LSTM+ benefits GPP prediction during periods that are affected by temporal patterns. For the temporal patterns evaluated, the LSTM<sub>perm</sub>+ and MLP+ behaved similarly, while the LSTM+ showed a performance benefit.

Furthermore, the LSTM<sub>perm</sub>+ had the same model capacity as the LSTM+. Combined, this indicates that the improved prediction of LSTM+ is due to a learned memory, compared to the LSTM<sub>perm</sub>+ and MLP+ which both only see contemporaneous predictor variables.

#### 4.4 Unknown effects of water stress are a dominating source of model error

We found that GPP can relatively reliably be predicted across relatively moist, winter-cold sites. For sites with a moisture index of 0.75 and above, the mean  $R^2$  for spatial out-of-sample predictions was 0.80. This indicates that – at least for the abiotic and biotic conditions represented in our dataset – GPP can be reliably simulated. Generalised models that spatially upscale yield relatively reliable results under such conditions with the  $R^2$  of spatial out-of-sample tests falling between 0.4 and 0.94.

However, under more arid conditions (MI <0.75) we found very variable performance of a generalised model. Different factors may cause poor generalisability across sites. Poor data quality with systematic differences of measurement errors across sites (Abramowitz et al., 2024), differences in functional relationships between GPP and its predictors across different species and vegetation types, or insufficient information in predictor variables all may underlie the variable performance of the global model across sites. Our results suggest that variable model performance is not clearly related to vegetation types (Figure 7). A tendency of poorer model performance in evergreen vegetation is likely related to limited information in remotely sensed greenness which is provided as a predictor (fAPAR). A clearer relationship of model performance was found across climate zones and across the gradient of the site's average aridity (Figure 7). Together with our finding of a clear relationship between the model prediction error and potential cumulative water deficits (Figure 4) and length of drought events (Figure 6), this suggests that poor model generalisability is linked to variable exposure and response to water stress across sites. Apparently, the history of precipitation and radiation is not sufficient to accurately model vegetation water stress exposure and responses, and effects on GPP.

Two factors are likely to undermine generalisability. First, responses to declining water potentials in the rooting zone are highly variable across species and linked to plant hydraulic traits and water use strategies. Even within broad classes of vegetation types, hydraulic relations of different plant species exhibit a wide variety (Choat et al., 2012; Joshi et al., 2022; Anderegg et al., 2018; Xu et al., 2016; Whitley et al., 2017; Konings and Gentine, 2017). Particularly in dry-adapted ecosystems (e.g., savannas and shrublands), contributions of different species to ecosystem-level integrated fluxes may also change over the season as a result of species-specific responses of leaf area to dryness (Xu et al., 2016; Whitley et al., 2017). Without related information provided to models, this complexity and the resulting variability of GPP responses to dryness cannot accurately be modelled across ecosystems with different species compositions.

Second, the exposure to water stress is highly variable across sites as a result of the surrounding topography and subsurface hydrology. Giardina et al. (2023) found strong variations of the functional relationship between evapotranspiration and cumulative water deficits, suggesting strongly variable rooting zone water storage capacities and plant access to groundwater across sites (Fan et al., 2017). Subsurface hydrology, groundwater influence, and belowground moisture convergence also appear to lead to large differences in ecosystem water balances at relatively dry sites (Hahm et al., 2019; McCormick et al., 2021). Several flux measurement sites have been identified as having greater mean annual precipitation than evapotranspiration, suggesting

subsurface moisture convergence and the influence of groundwater (Abramowitz et al., 2024). Due to the close link between evapotranspiration and GPP, these relations affect vegetation activity in general, including GPP. The surrounding topography, water holding capacity of the soil and weathered bedrock, groundwater table depth and rooting depth are either insufficiently known or specified from the predictors used for modelling GPP. Thus, related effects on GPP and the associated variability of water stress exposure cannot accurately be modelled across space. Variable water stress exposure and response affect GPP in relatively moist and energy-limited sites to a lesser degree than sites with frequent water limitation. Hence, a clear relation of model generalisability across aridity (Figure 8) emerges.

Another factor that contributes to these generalisation challenges is remotely sensed fAPAR, which is a predictor of GPP and tends to be less accurate in arid regions, where satellite sensors struggle to capture the large spatial heterogeneity at sub-pixel scales and may be affected by light absorption by non-photosynthetically active tissue (Kannenberg et al., 2024). In particular, noise due to sparsity, vegetation senescence and soil background lead to a frequent overestimation of fAPAR in arid and semiarid regions (Smith et al., 2019). This is exacerbated by the relative lack of ground observations in drylands, needed for calibration.

We expected that site-specific responses would be more effectively modelled by site-specific models compared to a generalised, global model. However, this was not unanimously the case. While the mean prediction error (RMSE) was generally lower for site-specific models, these models often predicted a smaller fraction of variation in the data than the global models. This was most clearly found for sites for which relatively short time series were available for model training. Hence, site-specific responses appear to be learnable, given sufficient data. Our interpretation is that  $R^2$  measures the ability to explain the variance in GPP, a task that becomes easier as the model sees more data. Put differently, a site-specific model can more accurately memorize the mean seasonal cycle of one particular site; but may not learn as well to deduce daily variations from observed changes in light and meteorology, due to its restricted sample. Indeed, sites where the global model was better (positive  $\Delta R^2$ ) invariably had relatively short observation periods (Figure 8), which increases the need to learn parts of the functional relation from other sites.

While the data-driven models showed advantages in certain situations with temporal effects, it is clear that challenges remain. Even with memory and accessible precipitation history, exposure and responses to water stress is not sufficiently modelled. This applies to models of GPP in general, including mechanistic models. As outlined above, it is likely that models need additional information to simulate GPP reliably at arid sites. Our experiments indicated that addition of commonly used satellite-derived variables boosts overall model performance without resolving the large variability in model performance across dry sites. This highlights the need for the inclusion of further information that is relevant for accurate drought response simulations. We expect that a similar reasoning and potential for model improvement applies also for mechanistic land surface models. The integration of additional satellite data to inform models for reliable simulations of water stress exposure in land surface models remains a largely unsolved task. Typically, land surface models account for water limitation by predicting leaf area index and stomatal conductance internally. Therefore, unknown exposure to water stress will likely remain a large source of uncertainty in land surface models. Similarly, mechanistic models like the P-model use fAPAR, but no other remote sensing data due to challenges in linking them in process representations.

#### 485 5 Conclusions

We have demonstrated that an LSTM – a popular type of recurrent deep neural network – is a powerful model to predict ecosystem GPP from local meteorological observations and remotely sensed fAPAR. Based on a spatial and temporal out-of-sample evaluation, we find that the model has significantly higher predictive skill than the theory-based P-model and outperforms a non-recurrent deep learning model under conditions of low root-zone moisture availability and very low temperatures in preceding weeks. The LSTM reliably simulates GPP dynamics across a range of environmental conditions and vegetation types (no agricultural vegetation tested here) at relatively moist sites (MI >0.75).

Through a detailed analysis of error patterns, we find that a recurrent model more accurately captures the GPP response to longer-term, cumulative impact. In particular, the LSTM adapts better to arid environments affected by water stress, a condition that builds up over time. Yet, we find that there is still a large variability in model skill across relatively arid sites, even if it outperforms both the mechanistic P-model and a memoryless neural network. The variability remains even with the inclusion of additional earth observation data, although this improves general model performance. This suggests that the model lacks information on variations in exposure and response to water stress and related effects on GPP. The inclusion of additional remotely-sensed and temporally varying information (e.g., sun-induced fluorescence (Li et al., 2018), vegetation optical depth (Konings and Gentine, 2017)) or static information about the topography, average groundwater table depth (Fan et al., 2013), and subsurface structure (Pelletier et al., 2016) as predictors for the deep learning models bears the potential for reducing errors and yielding more reliable GPP simulations in dry environments. As ecosystems are becoming more exposed to water limitation due to climate change (Denissen et al., 2022; Fu et al., 2024), it remains an important research topic to improve the predictability of ecosystem fluxes in the context of water stress.

Code and data availability. The code and data used in this study are available in the following GitHub repository: https://github.com/SamanthaBiegel/gpp-ml. Releases of this repository are archived on Zenodo (Biegel, 2025) The CSV file 'data/fdk\_v342\_ml.csv', which can be obtained from the repository, contains the dataset that is used as input to the machine learning experiments. The creation of this dataset can be reproduced with several steps. First, data is obtained from FluxDataKit v3.4.2 (Hufkens and Stocker, 2025), which gathers publicly accessible flux data from the major networks of eddy covariance sites described in section 2.1. The files from FluxDataKit are then processed using the script 'src/preprocess\_data.py', which results in the aforementioned CSV file. The data used as forcing for the P-model is available in a separate file in the repository that is derived from the CSV file and FluxDataKit metadata: 'R/drivers.rds'. Model experiments can be run with these two files as input by following the steps for environment preparation and experiment runs as detailed in the documentation page of the repository (https://github.com/SamanthaBiegel/gpp-ml). Predictions from all model experiments are stored in the directory 'preds/' and processed with 'figures.ipynb' to produce the figures presented here.

## Appendix A: Site information

| Sitename | Period    | MI   | Clim. | Veg. | Evergreen | Delayed GPP |
|----------|-----------|------|-------|------|-----------|-------------|
| AT-Neu   | 2002-2012 | 1.34 | Dfc   | GRA  | False     |             |
| AU-ASM   | 2012-2016 | 0.23 | BSh   | SAV  | False     |             |
| AU-Cum   | 2013-2018 | 0.60 | Cfa   | EBF  | True      |             |
| AU-DaS   | 2012-2017 | 0.74 | Aw    | SAV  | False     |             |
| AU-GWW   | 2013-2017 | 0.23 | BWh   | SAV  | False     |             |
| AU-Gin   | 2012-2017 | 0.42 | Csa   | WSA  | False     |             |
| AU-How   | 2009-2017 | 0.92 | Aw    | WSA  | False     |             |
| AU-Stp   | 2011-2016 | 0.50 | BSh   | GRA  | False     |             |
| AU-Tum   | 2011-2017 | 0.53 | Cfb   | EBF  | True      |             |
| AU-Ync   | 2012-2016 | 0.23 | BSk   | GRA  | False     |             |
| BE-Bra   | 2010-2020 | 1.17 | Cfb   | MF   | False     |             |
| BE-Dor   | 2011-2020 | 1.07 | Cfb   | GRA  | False     |             |
| BE-Maa   | 2016-2020 | 1.16 | Cfb   | CSH  | False     |             |
| BE-Vie   | 2000-2020 | 1.36 | Cfb   | MF   | False     | False       |
| CA-Ca1   | 2000-2009 | 3.03 | Cfb   | ENF  | True      |             |
| CA-Ca2   | 2001-2010 | 3.45 | Cfb   | ENF  | True      |             |
| CA-Gro   | 2004-2013 | 1.16 | Dfb   | MF   | False     |             |
| CA-Qfo   | 2004-2010 | 1.47 | Dfc   | ENF  | True      |             |
| CA-TP1   | 2009-2013 | 1.15 | Dfb   | ENF  | True      |             |
| CA-TP3   | 2008-2017 | 1.35 | Dfb   | ENF  | True      |             |
| CA-TPD   | 2012-2017 | 0.92 | Dfb   | DBF  | False     |             |
| CH-Aws   | 2015-2020 | 1.96 | ET    | GRA  | False     |             |
| CH-Cha   | 2010-2020 | 1.53 | Cfb   | GRA  | False     |             |
| CH-Dav   | 2000-2009 | 1.20 | ET    | ENF  | True      |             |
| CH-Fru   | 2011-2020 | 2.47 | Cfb   | GRA  | False     |             |
| CH-Lae   | 2005-2019 | 1.22 | Cfb   | MF   | False     |             |
| CH-Oe1   | 2003-2008 | 1.93 | Cfb   | GRA  | False     |             |
| CZ-BK1   | 2004-2019 | 1.88 | Dfb   | ENF  | True      |             |
| CZ-Lnz   | 2015-2020 | 0.66 | Dfb   | MF   | False     |             |
| CZ-RAJ   | 2012-2020 | 0.83 | Dfb   | ENF  | True      |             |
| CZ-Stn   | 2010-2020 | 0.96 | Dfb   | DBF  | False     |             |
| DE-Gri   | 2005-2019 | 1.38 | Cfb   | GRA  | False     |             |
| DE-Hai   | 2000-2019 | 1.22 | Cfb   | DBF  | False     | True        |
| DE-HoH   | 2015-2020 | 0.66 | Cfb   | DBF  | False     |             |
| DE-Obe   | 2009-2020 | 1.64 | Cfb   | ENF  | True      |             |

Continued on next page

| Sitename | Period    | MI   | Clim. | Veg. | Evergreen | Delayed GPP |
|----------|-----------|------|-------|------|-----------|-------------|
| DE-RuR   | 2012-2020 | 1.52 | Cfb   | GRA  | False     |             |
| DE-RuW   | 2013-2020 | 1.66 | Cfb   | ENF  | True      |             |
| DE-Tha   | 2000-2019 | 1.18 | Cfb   | ENF  | True      |             |
| DK-Sor   | 2000-2012 | 1.75 | Cfb   | DBF  | False     |             |
| ES-Abr   | 2016-2020 | 0.35 | Csa   | SAV  | False     |             |
| ES-Agu   | 2007-2013 | 0.28 | BSk   | OSH  | False     |             |
| ES-LJu   | 2006-2015 | 0.72 | Csa   | OSH  | False     |             |
| ES-LM1   | 2015-2020 | 0.54 | Csa   | SAV  | False     |             |
| ES-LM2   | 2015-2020 | 0.51 | Csa   | SAV  | False     |             |
| FI-Hyy   | 2000-2016 | 1.27 | Dfc   | ENF  | True      | False       |
| FI-Let   | 2010-2020 | 1.29 | Dfb   | ENF  | True      |             |
| FI-Sod   | 2008-2014 | 1.74 | Dfc   | ENF  | True      |             |
| FI-Var   | 2016-2020 | 1.70 | Dfc   | ENF  | True      |             |
| FR-Bil   | 2015-2020 | 0.99 | Cfb   | ENF  | True      |             |
| FR-FBn   | 2009-2020 | 0.63 | Csa   | MF   | False     |             |
| FR-Fon   | 2006-2013 | 0.96 | Cfb   | DBF  | False     |             |
| FR-LBr   | 2003-2008 | 1.04 | Cfb   | ENF  | True      |             |
| FR-Pue   | 2001-2013 | 1.00 | Csa   | EBF  | True      |             |
| IL-Yat   | 2012-2020 | 0.18 | BSh   | ENF  | True      |             |
| IT-Col   | 2007-2014 | 1.29 | Cfa   | DBF  | False     |             |
| IT-Cpz   | 2001-2007 | 0.65 | Csa   | EBF  | True      |             |
| IT-Lav   | 2003-2020 | 1.41 | Cfb   | ENF  | True      |             |
| IT-Lsn   | 2016-2020 | 1.12 | Cfa   | OSH  | False     |             |
| IT-MBo   | 2004-2012 | 1.88 | Dfb   | GRA  | False     |             |
| IT-Ren   | 2001-2014 | 1.27 | Dfc   | ENF  | True      |             |
| IT-Ro1   | 2002-2006 | 0.84 | Csa   | DBF  | False     |             |
| IT-Ro2   | 2002-2007 | 0.77 | Csa   | DBF  | False     |             |
| IT-Tor   | 2009-2020 | 1.94 | Dfc   | GRA  | False     |             |
| NL-Loo   | 2000-2017 | 1.07 | Cfb   | ENF  | True      | False       |
| RU-Fy2   | 2016-2020 | 0.96 | Dfb   | ENF  | True      |             |
| RU-Fyo   | 2000-2009 | 0.93 | Dfb   | ENF  | True      | False       |
| SE-Htm   | 2015-2020 | 1.30 | Cfb   | ENF  | True      |             |
| SE-Nor   | 2014-2020 | 0.89 | Dfb   | ENF  | True      |             |
| SE-Ros   | 2015-2020 | 1.59 | Dfc   | ENF  | True      |             |
| US-BZS   | 2016-2020 | 0.75 | Dfd   | ENF  | True      |             |

Continued on next page

| Sitename | Period    | MI   | Clim. | Veg. | Evergreen | Delayed GPP |
|----------|-----------|------|-------|------|-----------|-------------|
| US-Bar   | 2005-2017 | 1.54 | Dfb   | DBF  | False     |             |
| US-Blo   | 2001-2006 | 1.19 | Csb   | ENF  | True      |             |
| US-Fmf   | 2006-2010 | 0.51 | Csb   | ENF  | True      |             |
| US-GLE   | 2006-2019 | 1.73 | Dfc   | ENF  | True      |             |
| US-Ha1   | 2000-2020 | 0.80 | Dfb   | DBF  | False     | True        |
| US-Ho2   | 2007-2017 | 1.06 | Dfb   | ENF  | True      |             |
| US-ICh   | 2010-2021 | 1.60 | ET    | OSH  | False     |             |
| US-ICt   | 2016-2020 | 1.31 | ET    | OSH  | False     |             |
| US-Jo2   | 2011-2020 | 0.23 | BWk   | OSH  | False     |             |
| US-KFS   | 2008-2019 | 0.66 | Cfa   | GRA  | False     |             |
| US-KLS   | 2013-2019 | 0.42 | Cfa   | GRA  | False     |             |
| US-MMS   | 2000-2020 | 0.58 | Cfa   | DBF  | False     | True        |
| US-MOz   | 2007-2019 | 0.86 | Cfa   | DBF  | False     |             |
| US-Me2   | 2005-2010 | 0.64 | Csb   | ENF  | True      |             |
| US-Mpj   | 2009-2020 | 0.30 | BSk   | WSA  | False     |             |
| US-NR1   | 2000-2015 | 0.67 | Dfc   | ENF  | True      |             |
| US-PFa   | 2000-2014 | 0.52 | Dfb   | MF   | False     | True        |
| US-Rms   | 2015-2019 | 0.51 | BSh   | CSH  | False     |             |
| US-Ro4   | 2015-2021 | 1.28 | Dfa   | GRA  | False     |             |
| US-Rwf   | 2015-2019 | 0.59 | BSh   | CSH  | False     |             |
| US-Rws   | 2015-2019 | 0.69 | BSk   | OSH  | False     |             |
| US-SRG   | 2009-2014 | 0.37 | BSk   | GRA  | False     |             |
| US-SRM   | 2005-2014 | 0.29 | BSk   | WSA  | False     |             |
| US-Seg   | 2007-2021 | 0.27 | BSk   | GRA  | False     |             |
| US-Ses   | 2008-2021 | 0.24 | BSk   | OSH  | False     |             |
| US-Syv   | 2002-2006 | 1.14 | Dfb   | MF   | False     |             |
| US-Ton   | 2002-2014 | 0.50 | Csa   | WSA  | False     |             |
| US-UMB   | 2000-2014 | 0.41 | Dfb   | DBF  | False     |             |
| US-UMd   | 2008-2021 | 1.21 | Dfb   | DBF  | False     |             |
| US-Var   | 2001-2020 | 0.65 | Csa   | GRA  | False     |             |
| US-WCr   | 2000-2005 | 1.08 | Dfb   | DBF  | False     |             |
| US-Whs   | 2009-2015 | 0.26 | BSk   | OSH  | False     |             |
| US-Wjs   | 2008-2021 | 0.29 | BSk   | SAV  | False     |             |
| US-Wkg   | 2005-2021 | 0.31 | BSk   | GRA  | False     |             |

Table B1. Overview of the data sources

| Variable                      | Description                            | Source                                      |
|-------------------------------|----------------------------------------|---------------------------------------------|
| Main experiments (all models) |                                        |                                             |
| GPP                           | Target variable (GPP_NT_VUT_REF)       | PLUMBER2 (flux sites)                       |
| Meteorology                   | TA_F_MDS, TA_DAY_F_MDS, SW_IN_F_MDS,   | Flux sites                                  |
|                               | LW_IN_F_MDS, VPD_DAY_F_MDS, PA_F, P_F, |                                             |
|                               | WS_F                                   |                                             |
| fAPAR                         | Absorbed solar radiation               | MODIS MCD15A2H (500 m, 8-day)               |
| MODIS reflectance bands + LST | RED, NIR, BLUE, GREEN, SWIR1,          | FluxnetEO                                   |
|                               | SWIR2, SWIR3, LST_TERRA_Day_VZA0,      |                                             |
|                               | LST_TERRA_Night_VZA0                   |                                             |
| Additional predictors         |                                        |                                             |
| Soil moisture                 | Modelled soil water availability       | SPLASH water balance model                  |
| Water holding capacity        | Root zone storage                      | Stocker et al. (2023)                       |
| Site features                 |                                        |                                             |
| Moisture index (MI)           | P/PET                                  | SPLASH model + site precipitation           |
| Vegetation type               | IGBP                                   | International Geosphere-Biosphere Programme |
| Climate zone                  | Köppen-Geiger climate zone             | Beck et al. (2018)                          |

**Table B2.** Overview of the implemented models.

| Model            | Key components                                             | Implementation |
|------------------|------------------------------------------------------------|----------------|
| LSTM             | LSTM cells with LayerNorm + linear layers (until 16 units) | PyTorch        |
| MLP              | Linear layers only (until 16 units)                        | PyTorch        |
| $LSTM_{perm} \\$ | Same as LSTM, permuted input                               | PyTorch        |
| P-model          | FvCB photosynthesis, soil moisture stress                  | R (rsofun)     |

## Appendix B: Methods details

## **Appendix C: Sequence models**

520 We evaluated two additional sequence models using the global model setup.

**Temporal Convolutional Network.** We implemented a Temporal Convolutional Network (TCN) following Bai et al. (2018). Similar to the LSTM, we used GELU activations (Hendrycks and Gimpel, 2023) and LayerNorm (Ba et al., 2016).

**LSTM+attention**. We also implemented a second version of the LSTM that uses a multi-head (8) attention mechanism (Vaswani et al., 2017) with two linear layers after the LSTM layer.

Table B3. Cross-validation and training setups.

|                | Global model (Spatial CV)                | Site-specific model (Temporal CV)                |  |
|----------------|------------------------------------------|--------------------------------------------------|--|
| Data split     | 5 folds, stratified by mean TA_F_MDS, MI | Years per site, started at coldest/wettest month |  |
| Inner CV       | 3-fold (20 configurations)               | 5 held-out years (40 configurations)             |  |
| Training size  | 128 days, 32 days overlap                | 128 days, 32 days overlap                        |  |
| Early stopping | Max 50 epochs, patience 10               | Max 50 epochs, patience 10                       |  |
| Evaluation     | $R^2$ , RMSE, bias, % error              | $R^2$ , RMSE, bias, % error                      |  |

Figure C1. Comparison of the LSTM against the TCN (a) and against the LSTM+attention (b).  $R^2$  of predicted versus observed values of daily GPP per site from the global cross-validation is shown for the LSTM along the y-axis and the TCN/LSTM+attention along the x-axis. The dotted line indicates equal performance.

## Appendix D: Additional figures

525

*Author contributions.* SB implemented the methods and wrote the first draft of the paper. BDS processed part of the eddy covariance data. All authors developed the study, discussed the analyses and contributed to writing the paper.

Competing interests. The authors declare no competing interests.

**Figure C2.** The model bias per day before and after the start of drought events for three sequence models. The bias is calculated as the difference between model predictions and observations. The bias is normalised with respect to the median value during the 1 to 10 days before each event onset. Shaded areas mark the area between the 33% and 66% quartiles.

**Figure D1.** Number of sites as well as sequence lengths per site for different levels of the moisture index. The number of sites are counted per bin of width 0.2 of the moisture index.

Acknowledgements. We thank Piersilvio De Bartolomeis, Alexandru Meterez, Zixin Shu and Josefa Arán Paredes for their assistance with early exploratory work and code development that provided input to the development of this work. Samantha Biegel acknowledges support from the ETH AI Center.

**Figure D2.** The distribution of performance metrics for sites with a moisture index below 0.75, and sites with a moisture index above 0.75. The individual values as well as box plots are shown.

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
