# Peer review of "Unrecognised water limitation is a main source of uncertainty for models of terrestrial photosynthesis"

_EGUsphere, 2025_

## Author Response (AR1)

**Response to Reviewer 1**

Legend

Black text: reviewer comments

Blue text: original responses to comments Red text: description of changes to manuscript

https://doi.org/10.5194/egusphere-2025-1617-RC1

**General comments:**

To predict ecosystem gross primary productivity (GPP), this manuscript utilizes ecosystem flux data, meteorological measurements from 109 globally distributed sites, and remotely sensed vegetation indices to train three models: a mechanistic, theory-based photosynthesis model, a memoryless multilayer perceptron (MLP) and a recurrent neural network (Long Short-Term Memory, LSTM). The authors found that both deep learning models outperform the P-model, and the LSTM performs best. Particularly, model skill is consistently good across moist sites with strong seasonality. Model error tends to increase with increasing potential cumulative water deficits. The LSTM adapts better to arid environments affected by water stress, yet there is still a large variability in model skill across relatively arid sites.

This is an interesting analysis and the topic is pretty important. Overall, I find the paper compelling and fit for publication after revision. I include my comments below, which I hope help the authors to further strengthen the paper.

1. Some important details should be provided in this paper. Firstly, the methods and dataset for determining the optimal hyper-parameter of the proposed model are not provided.

Fair point. Our aim is certainly full reproducibility of all experiments, and we will make sure to clarify all remaining details, e.g., the hyper-parameter tuning approach, in the revised manuscript. We will add details on the number of data folds used, the stratification approach, and evaluation metric. Moreover, the code to reproduce the experiments has been published open-source.

We have expanded the description of the hyperparameter tuning approach:

L193-199: "In the spatial cross-validation setup, a 3-fold inner cross-validation was performed within each of the 5 data folds. The sites were distributed based on mean air temperature and moisture index to ensure equal representation of climate types across folds. The hyperparameters were tuned using random sampling of 20 configurations. In the temporal cross-validation setup, hyperparameters were tuned for each held-out test year based on the remaining

years of data for each site. A cross-validation with 5 randomly chosen held-out validation years was performed. In this setup, the hyperparameters were tuned using random sampling of 40 configurations. In both setups, the configuration with the lowest average validation RMSE across the held-out folds was selected."

Additionally, the rationale behind selecting these three specific models, especially the combination of machine learning models (LSTM and MLP) with the process-based P-model, should be better justified. It is also recommended to explore and compare other state-of-the-art models, such as gated recurrent units (GRU), convolutional neural networks (CNN), and other sequence modeling approaches, to provide a more comprehensive evaluation.

OK. While we do not expect big differences with alternative sequence models, we will add further comparisons in the appendix. We will include this in our justification of the temporal model choice. We will also further clarify the choice of the process-based model as a benchmark with a known treatment of temporal effects.

We have evaluated two additional sequence models: a Temporal Convolutional Network and LSTM with an attention layer. We have described these in Appendix C. We have included a comparison between sequence models in terms of overall performance as well as performance during drought events.

We have included references to this analysis in the Methods and Results sections:

L242-244: "Finally, we evaluated a Temporal Convolutional Network (Bai et al., 2018) and an LSTM with attention layer (Vaswani et al., 2017) as two alternative sequence models (Appendix C). We compared their overall performance as well as their event response performance to the LSTM."

L285-287: "Evaluations of the alternative sequence models (Appendix C) showed that while the TCN+ performed better at 60 out of 109 sites (Figure C1), no difference was found between models in terms of their performance during drought events (Figure C2), supporting the further evaluation of the LSTM+ for temporal patterns."

We have added a line in the introduction to support the choice of comparison with the P-model:

L90-91: "The P-model serves as a benchmark with known treatment of temporal effects."

2. In order to prove the superiority of the proposed models, this paper has carried out several forecasting experiments. Three models simulate GPP dynamics

across a range of environmental conditions and vegetation types, please give numbers of samples for each classification.

You are right, we overlooked this. We will annotate the figures with the number of sites in each category.

We have added site number annotations to the relevant figures (Figure 4, 5, and 7).

Furthermore, figure descriptions must be more precise. For example, in Section 3.3 (Lines 240-255), the discussion of Figure 6 should clearly indicate that it comprises two sub-figures and describe each accordingly.

Thank you, we will add more comprehensive descriptions to the affected figures. In general, we prefer not to overload figure captions, we hope it is ok not to repeat information that is already described in the text.

We have expanded the figure descriptions of Figure 7 (originally 6), Figure 8, and Figure 10.

3. Please make sure all figures are clear. The parameters and other details of the proposed model and methods should be organized in some tables.

We will double-check the clarity of the figures and will add tables with model details in the appendix.

We have added several tables with details of our methods. They can be found in Appendix B: table B1, B2 and B3. They are referenced in the text in the following location:

L245: "The details of the data, models and experimental setup are summarised in Table B1. Table B2 and Table B3."

**Minor Suggestions:**

 Please provide a description of the machine learning model construction, including the procedures for training and test dataset selection, normalization or preprocessing methods, and any other relevant implementation details necessary for reproducibility.

We will revise the experiments section to further clarify the details of the model construction and architecture: we divided all sites into 5 folds with stratified random sampling, then each fold in turn serves as the test set, with the four others as training set. A small subset of the training sites (20%) was set aside as validation set to enable early stopping of the training process and hyperparameter tuning.

We have added the following sentence:

L200-201: "After hyperparameter tuning, models were trained on the full data folds with the chosen hyperparameter configurations, with 20% of the sites used as validation data for early stopping of the training."

We have also checked for completeness regarding other model construction details.

2. In Figure 1, it is recommended to include the number of observation sites corresponding to each aridity type.

Thanks, we will add the number of sites in each category to the legend.

We have changed Figure 1 accordingly.

3. Methods requires citing references, please check.

We have added a reference for GELU activations (L145).

4. The explanation of part "3.3 Spatial patterns in model performance" (corresponding to Figure 6) is somewhat unclear. "Across sites with moisture index P/PET ≥ 0.75 the R₂ is 0.76, whereas it was only 0.57 for more arid sites (MI <0.75). The (normalised) RMSE follows a similar pattern, with a value of 0.88 for sites with MI <0.75, compared to 0.57 for moist sites." However, these values are not directly visible in the two subplots on the left panel of Figure 6.

These values are indeed not directly visible in the referenced figures: they are derived from aggregations of the sites belonging to the two different aridity levels, whereas the current figures only display the performances at each individual site. We will make this clearer by adding panels for the specific categorisation used here.

We have included a new figure in the Appendix showing how these numbers are derived from the data (Figure D2).

**Response to Reviewer 2**

Legend

Black text: reviewer comments
Blue text: responses to comments

Red text: description of changes to manuscript

**https://doi.org/10.5194/egusphere-2025-1617-RC2**

Biegel et al. (2025) compare the performance of three modeling approaches for predicting gross primary productivity (GPP): an optimality-based photosynthesis model (the P-model), a memoryless multilayer perceptron (MLP), and a recurrent neural network (RNN) with long short-term memory (LSTM) that incorporates memory of past environmental conditions. The models are evaluated in two settings: (1) site-level training and evaluation to assess temporal dynamics, and (2) cross-site evaluation to assess spatial generalization. The authors report that both MLP and LSTM outperform the P-model, with LSTM showing the greatest advantage under dry conditions due to its ability to account for temporal memory. However, they also note that all models struggle to accurately simulate GPP at certain dry sites, likely due to limited representation of water stress and its impact on carbon fluxes.

While the modeling framework and experimental setup are generally sound and the results compelling, the manuscript revisits a well-studied topic in the eco-hydrological machine learning literature. Numerous recent studies have explored the use of RNNs and LSTMs to simulate ecological processes and quantify memory effects (e.g., Montero et al., 2024; Agarwal et al., 2023; Cattry et al., 2025; Kraft et al., 2019, 2021; Wesselkamp et al., 2025; Zhao et al., 2025). Many of these works directly compare memory-based RNNs to memoryless architectures in the context of simulating vegetation states or fluxes, often using methodologies closely aligned with the current study. This overlap does not diminish the potential contribution of Biegel et al., but it raises the bar for novelty and interpretability. Unfortunately, that potential remains underdeveloped in the current version of the manuscript.

For instance, in Figure 4, the authors compare absolute percentage errors of GPP predictions by aggregating results across PCWD bins. While this illustrates that LSTM and MLP models outperform the P-model under drier conditions, it does not clearly demonstrate that the LSTM's advantage stems specifically from its capacity to leverage temporal dependencies—such as drought memory or cold acclimation—rather than simply from its increased architectural complexity.

These are important points, we appreciate the suggestions for strengthening our arguments outlined in the review. Related work indeed provides comparisons of memory-based RNNs to memoryless architectures, therefore our goal was not to

provide another generic comparison, but rather to dive deeper into when and why temporal modelling makes a difference. To this end, we regard the temporal model as an upper bound that takes into account the temporal dependencies within the available data. This helps us to gain new insights, by identifying limitations of model generalisation and the corresponding environmental factors. As outlined in the responses below, we will add several additional experiments to further establish the connection between temporal GPP patterns and the memory mechanism of the LSTM.

To strengthen this argument, the authors could analyze model performance during known extreme events (e.g., multi-week droughts or cold spells) and assess whether LSTM models exhibit better generalization or resilience.

This would be a valuable addition to the experiments based on cumulative water deficit and seasonal cycles. We will include an evaluation of model performance during drought events and frost events. Our plan is to identify drought events based on daily estimates of anomalies in light use efficiency due to soil moisture (<a href="https://zenodo.org/records/1158524">https://zenodo.org/records/1158524</a>), following Stocker et al. (2018). Frost events will be determined based on air temperature. We believe that an evaluation of extreme events derived from meteorological data is more meaningful than using individual "known" events whose definition is to some degree subjective.

Stocker, Benjamin D., Jakob Zscheischler, Trevor F. Keenan, I. Colin Prentice, Josep Peñuelas, and Sonia I. Seneviratne. 2018. 'Quantifying Soil Moisture Impacts on Light Use Efficiency across Biomes'. New Phytologist 218 (4): 1430–49. https://doi.org/10.1111/nph.15123.

We have included a new figure comparing models during drought events from the dataset as described above (Figure 6). We did not include a frost event analysis as we could not determine a clear signal from the currently available data.

We have included references to this new analysis in several sections:

L218-224: "We evaluated the temporal effect from soil moisture from a different angle through an analysis of model bias during drought events identified from the full dataset. Drought events were collected through the identification of sites and days where light use efficiency was reduced for at least 3 days (Stocker et al., 2018). All events were aligned at the onset. The bias was calculated as the difference between model predictions and observations and aggregated per day for the period between 10 days before the start of each event and up until 100 days after the start of each event. The bias values were normalised by subtracting each value by the median value during the window covering between 1 and 10 days before each event."

L283-284: "The event analysis (Figure 6) showed that the deep learning models predicted the drought response equally well during the first 20 days after an event. The LSTM+ predicted the response best from 20 days after the start of a drought event."

L377-381: "Event analysis also showed that the recurrent architecture helps predict the response in GPP to drought events (Figure 6). The benefit was observed primarily from 20 days after the start of a drought event. The similarity in prediction errors among models during the start of drought events and at lower levels of PCWD appears to indicate that the recurrent architecture only benefits the prediction of GPP at higher levels of water stress."

Additionally, applying permutation-based methods (as in Kraft et al., 2019) or interpretable machine learning tools (e.g., Integrated Gradients, as in Zhao et al., 2025) could help identify which temporal features or variables most strongly drive model predictions under different environmental conditions.

A permutation-based method in particular could help disentangle differences in model complexity and numerics from the ability to exploit temporal patterns. Therefore, we will evaluate another version of the LSTM trained with randomly shuffled input. While this still gives the LSTM some context about each site, it removes the chronological ordering needed to understand effects due to the meteorological history.

We have included a new version of the LSTM, called LSTMperm. This is the same model as the LSTM, but trained with randomly shuffled sequences of the data per site. We have added evaluations of this model to Table 2 (overall performance), Figure 3, Figure 4, Figure 5 and Figure 6. This includes all experiments done to assess temporal patterns in model performance.

We have included a reference to this model version in all sections of the text where the models are compared against each other in the context of these figures. In addition, we commented more extensively on this evaluation on these sections of the manuscript:

L13-15: "During periods affected by temporal patterns such as drought and frost, the LSTM shows lower model error than the MLP as well as an LSTM with shuffled input, showing that there is an advantage from learned temporal dependencies."

L151-154: "We also evaluated a version of the LSTM where we permute the temporal ordering (LSTM $_{perm}$ ). This model has the same architecture and capacity but it cannot rely on memory as the temporal patterns are eliminated from the input data. This version serves as an additional evaluation that disentangles the influence of the memory of the LSTM from other model properties such as the overall capacity."

L266-269: "From the overall performance metrics, it appeared that the LSTM+ is not clearly better than the MLP+, despite its ability to learn temporal patterns. However, the

improved performance of the LSTM+ compared to the LSTMperm+ indicates that temporal dependencies were learned. Targeted evaluations of temporal patterns give a clearer insight into the differences between the MLP+, the LSTMperm+ and the LSTM+."

L414-418: "Overall, we found that the LSTM+ benefits GPP prediction during periods that are affected by temporal patterns. For the temporal patterns evaluated, the  $LSTM_{perm}+$  and MLP+ behaved similarly, while the LSTM+ showed a performance benefit. Furthermore, the  $LSTM_{perm}+$  had the same model capacity as the LSTM+. Combined, this indicates that the improved prediction of LSTM+ is due to a learned memory, compared to the  $LSTM_{perm}+$  and MLP+ which both only see contemporaneous predictor variables."

The manuscript also includes a site-level versus global model comparison, which is a valuable angle. However, the practical implications of the observed performance gaps remain unclear. Such gaps could arise from various sources, including uneven spatial representation of ecosystems, differences in training data length, observational uncertainties, or intrinsic ecological variability. While Figure 8 suggests moisture index differences might explain some of the discrepancies, it is not clear whether these reflect genuine ecological signals or merely data sampling artifacts. These challenges—especially related to data imbalance and representativeness—are long-standing limitations in upscaling efforts like FLUXCOM and FLUXCOM-X. The authors could add considerable value by disentangling these effects and explicitly attributing performance gaps to either ecological complexity or sampling limitations.

These are indeed important points, and we carefully considered them during the development of the manuscript. In the experimental setup, we ensured even representation of sites with lower and higher values of the moisture index and lower and higher values of mean air temperature when splitting sites into training and evaluation groups. We also observed that across different moisture levels, the number of sites and the length of observations per site are distributed evenly (Figure 6 and 7). In particular, relatively arid sites are well-represented in our dataset. We will include an additional figure to display this more clearly.

Taken together, we concluded that uneven spatial representation, data sampling artifacts and differences in training data length are unlikely to explain the observed patterns and performance gaps with respect to aridity. Hence, we attributed them to varying exposure and response to water stress, an ecological factor that is insufficiently resolved by our model. We have highlighted several factors that might contribute to this variability, such as differences in plant hydraulic traits and belowground moisture dynamics. We will revise the discussion to clarify the link between these explanations and how we derived these from our attribution of performance gaps to ecological variability.

We have added a figure to the Appendix that shows the number of sites and length of observations per site at different aridity levels (Figure D1).

We have added further clarification to Discussion section 4.2 (now 4.3):

L403-407: "These patterns are unlikely to stem from uneven spatial representation, data sampling artifacts or differences in training data length. These were accounted for by ensuring even representation and sequence length of moist and arid sites (Figure D1) as well as even distribution of moisture index values when splitting sites during training. Therefore, we attribute performance gaps to the ability of the models to account for the ecological complexity under arid conditions."

Moreover, the discussion around the mechanistic model (P-model) and its comparison to data-driven approaches could be expanded. For instance, while the manuscript notes that the P-model encodes "rigid functional dependencies," it was originally developed to reduce dependency on calibration by incorporating plant optimality principles. If the central conclusion of this study is that machine learning models (especially LSTMs) consistently outperform mechanistic models, then the manuscript should provide a deeper reflection on how data-driven insights might inform or improve process-based models. Could the identified memory effects or variable sensitivities be translated into new empirical or semi-mechanistic formulations? What implications do the observed model deficiencies under water stress have for future land surface model development?

While we noted that our empirical results (in line with others) suggest that machine learning models outperform the mechanistic model, we do not intend this to be the central message of the study. The role of the mechanistic model in our evaluation is primarily to provide a benchmark where we know how it treats temporal effects. As shown in the analyses, the machine learning models show an advantage in certain situations that were selected based on current process understanding. However, we also highlighted the present limitations of data-driven modelling in these situations, with or without memory. Even when the precipitation history is accessible, we found that water stress is not sufficiently modelled, and we argue that a likely cause is a lack of information about variations in water stress response - a point that applies to GPP models in general, be they mechanistic or data-driven. One of our messages is that models cannot be expected to reliably simulate GPP at arid sites without that additional information - at least that is what current, necessarily incomplete evidence suggests.

There is potential for providing additional information from earth observation to the data-driven models. For example, relevant data may come from thermal remote sensing, microwave remote sensing, or multispectral reflectance data. For land surface models, integrating such data remains a largely unsolved task. Instead, these models typically predict LAI and stomatal conductance (the two key variables that determine

GPP under water limitation) internally. Therefore, unknown exposure to water stress will remain challenging to resolve and will likely remain a main source of uncertainty in land surface models.

Similarly, mechanistic models based on satellite data, i.e., light use efficiency models (e.g., MODIS, P-model), rely on fAPAR, whereas other remotely sensed observations are not used due to challenges in linking them to process representations in mechanistic models.

We will complement the revised manuscript with this discussion for further context. Additionally, to obtain further evidence to what extent GPP simulations might be affected by unrecognized water limitation, we will include an LSTM model with additional remote sensing data that is readily available.

We have complemented the discussion with a paragraph in Section 4.3 (now 4.4):

L473-484: "While the data-driven models showed advantages in certain situations with temporal effects, it is clear that challenges remain. Even with memory and accessible precipitation history, exposure and responses to water stress is not sufficiently modelled. This applies to models of GPP in general, including mechanistic models. As outlined above, it is likely that models need additional information to simulate GPP reliably at arid sites. Our experiments indicated that addition of commonly used satellite-derived variables boosts overall model performance without resolving the large variability in model performance across dry sites. This highlights the need for the inclusion of further information that is relevant for accurate drought response simulations. We expect that a similar reasoning and potential for model improvement applies also for mechanistic land surface models. The integration of additional satellite data to inform models for reliable simulations of water stress exposure in land surface models remains a largely unsolved task. Typically, land surface models account for water limitation by predicting leaf area index and stomatal conductance internally. Therefore, unknown exposure to water stress will likely remain a large source of uncertainty in land surface models. Similarly, mechanistic models like the P-model use fAPAR, but no other remote sensing data due to challenges in linking them in process representations."

We have evaluated new versions of all data-driven model setups with additional earth observation data from MODIS (Land Surface Temperature and reflectance bands). These are denoted with a plus in the model names. As these new versions showed significant improvements, we have replaced the original model predictions. We have kept the original versions in the overall model performance table (Table 2) to show the difference for each model at different aggregations. Due to the change in model predictions, many of the numbers reported in the Results section have been adjusted slightly. While the additional variables improved overall performance, they did not

change the performance differences and patterns observed in our experiments, which we believe supports the robustness of the observations discussed in our manuscript. We have referenced to the addition of MODIS variables here:

L17-18: "This was not resolved by the inclusion of additional earth observation data, although this improved overall model performance."

L77-85: "While the P-model only uses a single greenness index derived from remote sensing, previous work on flux modelling has shown that additional remotely sensed variables can be informative (Nelson et al., 2024; Kraft et al., 2024). The signal from thermal remote sensing may reflect changes in photosynthesis that are driven by physiological responses and stomatal regulation, affecting transpiration and therefore surface energy partitioning and surface heating. Therefore, land surface temperature (LST) may be useful information for GPP prediction. Common mechanistic and light use efficiency models don't consider this information as additional forcing. Furthermore, the full information of surface reflectance in all available individual bands may contain additional information about GPP changes as pigments deployed under stress conditions or the leaf water content can affect surface reflectance beyond what commonly used single greenness-based indices reflect (Ceccato et al., 2001; Gamon et al., 2016)."

L129-134: "We retrieved additional pre-processed remote sensing variables from the MODIS instruments through the FluxnetEO dataset (Walther et al., 2022). We included the first seven reflectance bands (RED, NIR, BLUE, GREEN, SWIR1, SWIR2, SWIR3) as well as daytime and nighttime land surface temperature (LST) at a viewing zenith angle of 0° (LST\_TERRA\_Day\_VZA0, LST\_TERRA\_Night\_VZA0).

The collection of flux sites was further filtered to include only sites with data available from the FluxnetEO dataset. This process resulted in a collection of 104 sites (Figure 1) with a total of 1020 site-years of data."

L235-238: "We evaluated two different sets of predictor variables. The standard set included the site-level meteorological measurements as well as fAPAR. The expanded set included LST variables and the MODIS reflectance bands. The model evaluations with this expanded set are denoted by a + after the model name. We compared this setup to the standard predictor set in terms of overall performance."

L249-250: "Furthermore, the model variants with the additional remote sensing variables (+ version) performed better than the models with only the standard predictor set."

L258-261: "While the LSTM+ clearly outperformed the LSTM in terms of overall metrics, the performance didn't improve at every site. At 68 out of 104 sites, the LSTM+

outperformed the LSTM. In further analyses, we focus on the + versions of each model, as the overall performance evaluation indicates that they are better predictors of GPP."

L476-478: "Our experiments indicated that addition of commonly used satellite-derived variables boosts overall model performance without resolving the large variability in model performance across dry sites. This highlights the need for the inclusion of further information that is relevant to drought response."

L495-496: "The variability remains even with the inclusion of additional earth observation data, although this improves general model performance."

In its current form, the manuscript presents an interesting comparison of modeling strategies but falls short of offering novel insights beyond existing literature. To warrant publication, the authors should:

- More rigorously establish the connection between memory mechanisms and improved performance under specific environmental stressors.
- Employ interpretable ML tools or event-based analyses to strengthen claims regarding temporal information use.
- Clarify the ecological or data-driven reasons behind site-global model performance differences.
- Reflect more deeply on how ML-based findings can inform mechanistic modeling efforts.

Addressing these points will substantially enhance the originality and relevance of the manuscript.

We appreciate the useful comments and suggestions from the reviewer and will take these on board for our revised manuscript. Our revision will, in particular, extend the discussion of our primary findings about data-driven modelling of temporal patterns in water-limited conditions. To summarize, we will implement the following analyses and discussions suggested by the reviewer:

- An additional evaluation of the LSTM, trained with permuted input. This shall strengthen the argument about the role of the memory mechanism to account for historical effects.
- An analysis of model performance during drought events and cold spells.
- Further discussion of the connection between limitations of data-driven models, mechanistic models and land surface models.
- An evaluation of the LSTM trained with additional remote sensing-derived variables.
- Further clarification and supporting information about our dataset, to better justify (i) the attribution of site-specific behaviour to ecological variability, and (ii) the implications for flux modelling.

The proposed changes have been implemented as outlined in the responses above.

**References:**

Cattry, M., Zhao, W., Nathaniel, J., Qiu, J., Zhang, Y., & Gentine, P. (2025). EcoPro-LSTM v0: A Memory-based Machine Learning Approach to Predicting Ecosystem Dynamics across Time Scales in Mediterranean Environments. *EGUsphere*, 2025, 1-37.

Kraft, B., Jung, M., Körner, M., Requena Mesa, C., Cortés, J., & Reichstein, M. (2019). Identifying dynamic memory effects on vegetation state using recurrent neural networks. *Frontiers in big Data*, *2*, 31.

Kraft, B., Besnard, S., & Koirala, S. (2021). Emulating ecological memory with recurrent neural networks. *Deep Learning for the Earth Sciences: A Comprehensive Approach to Remote Sensing, Climate Science, and Geosciences*, 269-281.

Wesselkamp, M., Chantry, M., Pinnington, E., Choulga, M., Boussetta, S., Kalweit, M., ... & Balsamo, G. (2025). Advances in land surface forecasting: a comparison of LSTM, gradient boosting, and feed-forward neural networks as prognostic state emulators in a case study with ecLand. *Geoscientific Model Development*, 18(4), 921-937.

Zhao, W., Winkler, A., Reichstein, M., Orth, R., & Gentine, P. (2025). Learning evaporative fraction with memory.